# Effect of long-term use of treated wastewater and clean water mixtures on soil heavy metal accumulation: An assessment using pollution indices

Hasan Er  *

Department of Biosystems Engineering, Faculty of Agriculture, Bingol University, Bingöl, Türkiye

* hasaner@bingol.edu.tr

**Citation:** Er H (2025) Effect of long-term use of treated wastewater and clean water mixtures on soil heavy metal accumulation: An assessment using pollution indices. PLoS One 20(12): e0337424. https://doi.org/10.1371/journal.pone.0337424

## Abstract

The growing scarcity of clean water resources has driven increased interest in the use of treated wastewater (WW) as an alternative source for agricultural irrigation. While WW provides essential nutrients that can support plant growth and soil fertility, its long-term application may lead to the accumulation of heavy metals in soil, posing environmental and health risks. This study aimed to assess the effects of a four-year field experiment using different ratios of treated wastewater (WW) and clean water (CW) mixtures on soil heavy metal accumulation. Additionally, these effects were evaluated using widely accepted pollution indices, including the Contamination Factor (CF), Enrichment Factor (EF), Geoaccumulation Index (Igeo), and Pollution Load Index (PLI). A four-year field experiment was conducted using five irrigation regimes: W1 (100% WW), W2 (75% WW + 25% CW), W3 (50% WW + 50% CW), W4 (25% WW + 75% CW), and W5 (100% CW as control). The accumulation of Cr, Ni, Cu, Zn, Cd, Fe, and Pb was monitored. Results revealed that heavy metal concentrations increased with higher WW proportions and longer irrigation duration, but all levels remained below national and international regulatory limits. CF values approached moderate contamination levels for Pb, Fe, Zn, and Cd under W1, while Cr, Ni, and Cu indicated low contamination potential. EF values were mostly below 1.0, especially under mixed irrigation schemes, suggesting minimal anthropogenic enrichment. Igeo values were negative across all treatments, with the lowest values in the control group, classifying the soils as unpolluted. PLI values were consistently below 1.0, indicating an overall unpolluted status. The findings suggest that moderate blending ratios (25–50% WW) can limit heavy metal buildup while maintaining soil productivity. Therefore, the controlled use of treated wastewater in appropriate mixtures emerges as a promising approach for sustainable agricultural irrigation, balancing productivity with environmental protection.

**Data availability statement:** All datasets generated for this study can be found in Figshare (https://doi.org/10.6084/m9.figshare.30209449).

**Funding:** The author(s) received no specific funding for this work.

**Competing interests:** The authors have declared that no competing interests exist.

## Introduction

With the increasing world population, the need for water is also increasing, and this situation can reach dimensions that threaten human life in some regions [1,2]. Especially in recent years, drought and water scarcity on a global scale have become an important problem for governments, policy makers and water managers [3–5]. This has increased the use of non-conventional water sources such as urban and industrial wastewater [6]. Wastewater can have yield-enhancing effects in agriculture due to its high levels of organic matter and nutrients [7–9]. However, this practice, which has become widespread due to the lack of clean water, can also cause heavy metal accumulation in the soil [10,11]. Heavy metals have become important environmental pollutants with the increase in industrial and agricultural activities [12,13]. The accumulation of these pollutants can cause harmful effects on the environment and human health in the long term. Therefore, quantitative assessment of the impact of irrigation water quality on heavy metal accumulation in soil is of great importance. This assessment can be done in a systematic and comparable way through indices such as Contamination Factor (CF), Enrichment Factor (EF), Geoaccumulation Index (Igeo) and Pollution Load Index (PLI) [14–16]. Due to their persistent and non-degradable nature, heavy metals in soil can easily pass into plants and become part of the food chain [17,18]. This can negatively affect plant growth and yield, as well as pose a risk to human health [19]. Therefore, rather than using wastewater directly, mixing it with clean water and using it in irrigation is considered a safer alternative to limit heavy metal accumulation in soil. For example, [20] reported that wheat germination was highest under conditions where 25% wastewater was used. Other studies have similarly reported that the use of wastewater at controlled rates increases crop yields [21,22]. However, significant increases in heavy metal accumulation in soil have also been reported in long-term irrigation with wastewater [6,23]. Many studies have shown that soils irrigated with wastewater have higher heavy metal contents than those irrigated with clean water [24–26]. Therefore, regular monitoring and evaluation of heavy metals accumulated in soil is necessary to protect public health.

Nonetheless, the literature extensively examines the beneficial impacts of utilizing treated wastewater in agriculture on soil fertility, crop quality, and heavy metal buildup. Nevertheless, research regarding the impact of these techniques on the long-term deposition of heavy metals and the mitigation of such buildup through dilution with clean water at various ratios is somewhat few. While many studies have investigated the short-term consequences of wastewater irrigation, the long-term influence of varying ratios of treated wastewater to clean water on soil heavy metal buildup is little explored, especially when evaluated using various pollution indices. This study provides a unique addition by assessing the impact of treated wastewater on soil heavy metal buildup over four years in the semi-arid climate of Eastern Anatolia, Türkiye, despite similar studies completed in other places. Furthermore, the use of a gradient-based irrigation strategy with five distinct wastewater/clean water ratios (ranging from 100% wastewater to 100% clean water) allows for a more detailed analysis of the dose-dependent effects on soil pollution indices. To our knowledge,

few studies have addressed such long-term and systematically varied irrigation treatments in this regional context, making our findings particularly relevant for the sustainable management of water resources in water-scarce environments. The primary aim of this study is to investigate how 4-year irrigation using different ratios of treated wastewater (WW) and clean water (CW) affects the accumulation of heavy metals (Cr, Ni, Cu, Zn, Cd, Fe, and Pb) in agricultural soil. Specifically, five irrigation treatments—100% WW, 75% WW + 25% CW, 50% WW + 50% CW, 25% WW + 75% CW, and 100% CW (control)—were applied over four consecutive years. The accumulation levels were evaluated using four widely accepted pollution indices: Contamination Factor (CF), Enrichment Factor (EF), Geoaccumulation Index (Igeo), and Pollution Load Index (PLI). Additionally, the study aims to assess the environmental risks and potential benefits of reusing treated wastewater in agriculture, with a focus on sustainable soil management and safe irrigation practices.

## Materials and methods

### Experimental area and climate

Field studies were conducted in the experimental field of Bingöl University Faculty of Agriculture located at 38°33′ North latitude, 40°30′ East longitude and 1030 m altitude in Bingöl province, eastern Turkey, during the vegetation period from May to September of each year between 2021 and 2024 (Fig 1).

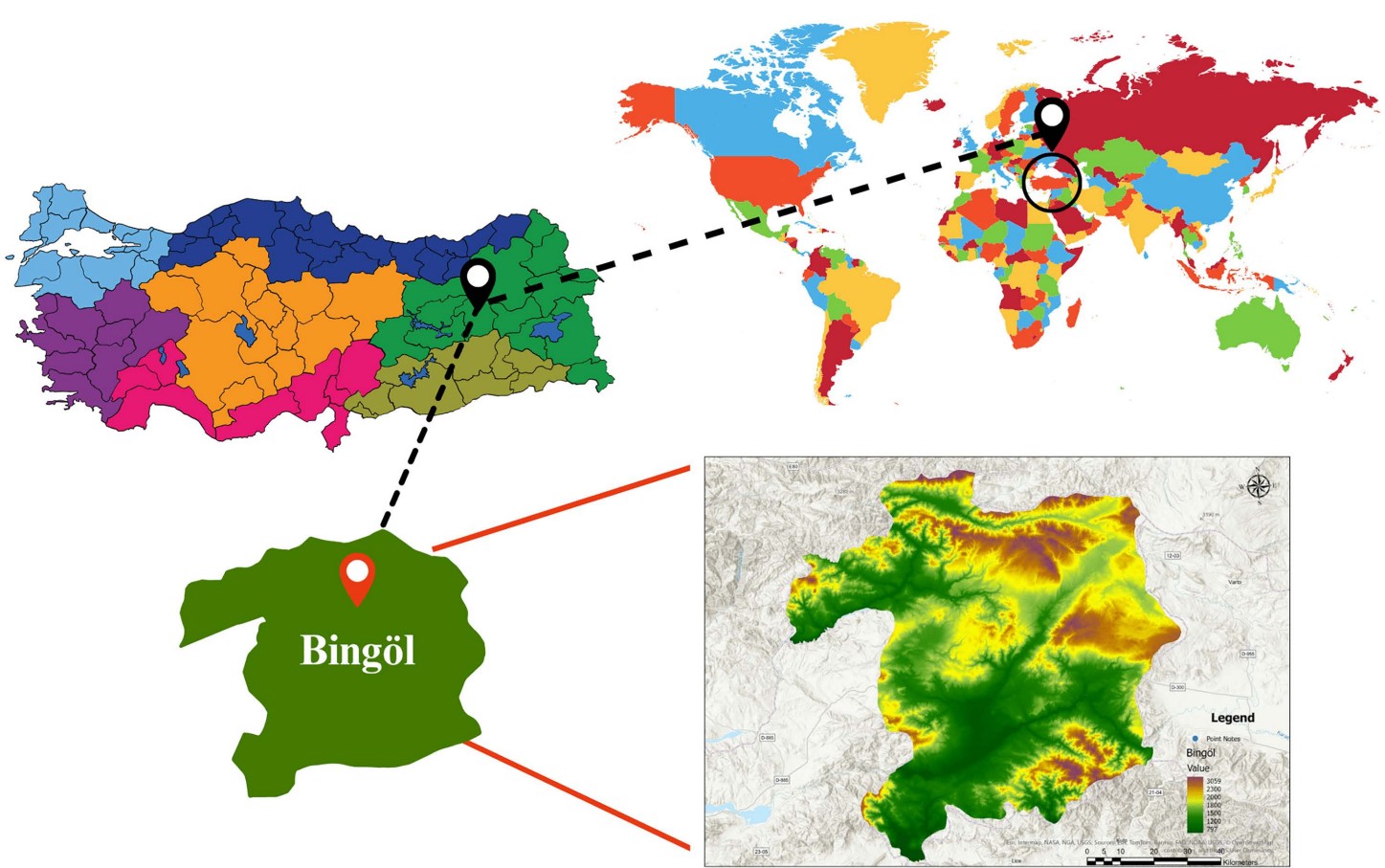

**Fig 1. Map of study area.**

According to the long term meteorological data for 2021–2024 and 1961–2024, May-September, which is considered as the vegetation period in Bingöl province, is a period in which continental climate characteristics are prominent. During this period, the summer months are quite hot and dry, and precipitation is generally concentrated at the beginning of the season (Fig 2). Mean monthly temperatures ranged from 21.3°C to 28.4°C, which is consistent with long-term averages (21.3°C–26.7°C), indicating no extreme temperature deviations that could have affected plant physiological responses or yield. Precipitation showed slight interannual variability, with total rainfall ranging between 6–40 mm in most years, compared to the long-term average of 49.2 mm. The years 2022 and 2023 received relatively higher rainfall early in the season, while 2021 and 2024 were drier, particularly in mid-summer. Relative humidity varied between 26.4% and 47.8%, slightly below the long-term average (36–44%), reflecting the arid conditions typical of the region. The analysis indicates that both precipitation and temperature trends during the study period closely followed the regional long-term pattern. Therefore, it is considered that the experimental results were not significantly affected by seasonal variability, as climatic conditions remained relatively stable across the study years.

## Soil and irrigation water characteristics

The soil in the research area has a clayey-loamy structure and does not have any problems in terms of salinity. Although the organic matter content is low, it is rich in phosphorus and potassium. The basic physical and chemical properties of the soil and irrigation water are presented in Table 1, respectively.

Tap water was preferred as a clean water source for irrigation. Treated wastewater was obtained from the discharge point of the Urban Waste Water Treatment Plant in Bingöl. Before irrigation, the water was transported to the experimental area and transferred to polyethylene tanks from where it was made available for use. During the irrigation period, water

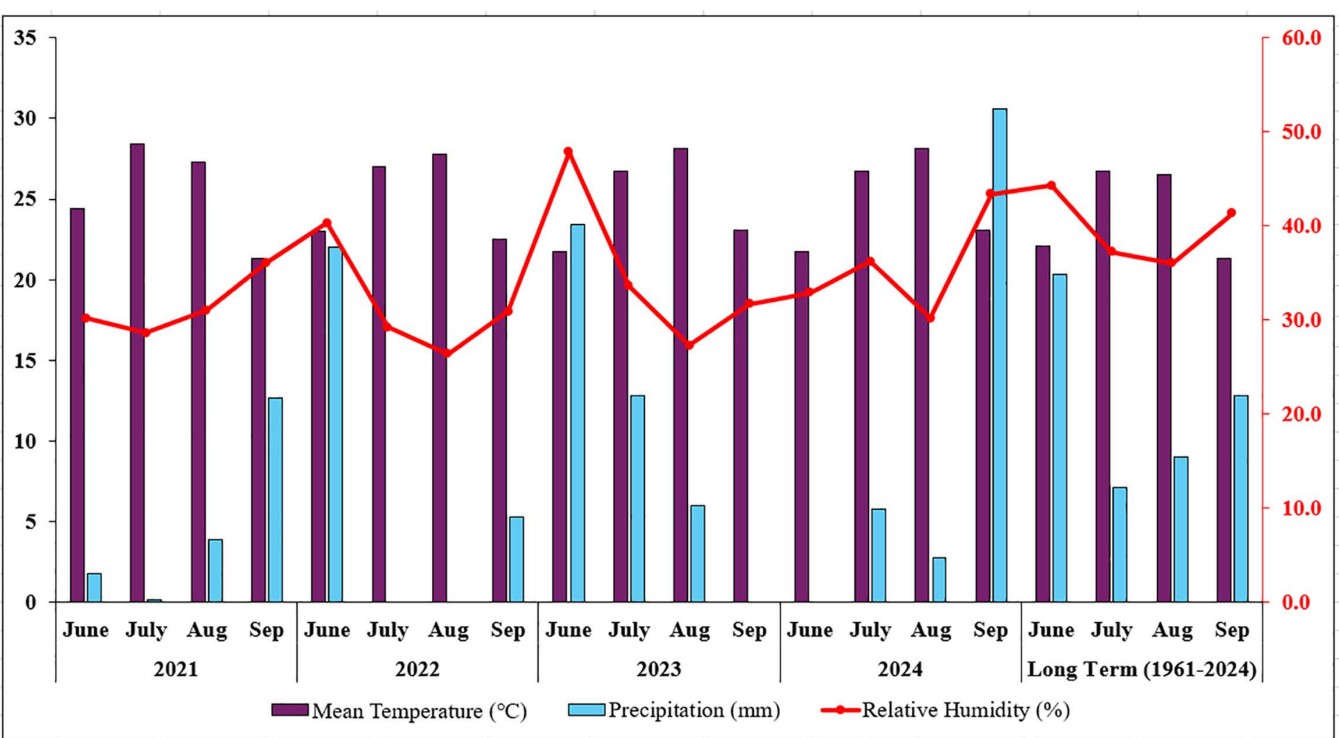

**Fig 2. Climate parameters of Bingöl during the vegetation period (2021–2024) and long-term (1961–2024).**

**Table 1. Some physical and chemical properties of the experimental soil.**

| Property | Value |
|---|---|
| Soil texture class | clay loam |
| Sand (%) | 30.6 |
| Silt (%) | 36.2 |
| Clay (%) | 33.2 |
| EC (dSm$^{-1}$) | 0.14 |
| pH | 6.01 |
| CaCO$_3$ (%) | 0.8 |
| Soil organic matter (%) | 0.51 |
| Total Kjeldahl N (%) | 0.03 |
| P$_2$O$_5$ (kg ha$^{-1}$) | 92 |
| K$_2$O (kg ha$^{-1}$) | 828 |
| Exchangeable Ca (cmol kg$^{-1}$) | 11.15 |
| Exchangeable Mg (cmol kg$^{-1}$) | 3.74 |
| Exchangeable K (cmol kg$^{-1}$) | 0.82 |
| Exchangeable Na (cmol kg$^{-1}$) | 0.21 |
| CEC (cmol kg$^{-1}$) | 15.92 |
| Extractable Sodium (Na) (cmol kg$^{-1}$) | 0.88 |
| ESP (%) | 5.5 |
| B (mg kg$^{-1}$) | 0.001 |
| Cu (mg kg$^{-1}$) | 0.68 |
| Mn (mg kg$^{-1}$) | 16.11 |
| Zn (mg kg$^{-1}$) | 1.68 |

EC: Electrical conductivity, CEC: cation exchange capacity, ESP: exchangeable sodium percentage

samples were taken regularly to represent each month and analyzed for quality. Chemical properties of irrigation water are given in Table 2.

It was determined that the quality parameters and heavy metal contents measured in irrigation water were below the limit values determined by the classification of inland water resources of the "Water Pollution ControlRegulation" [27], "Communiqué on Technical Procedures of Wastewater Treatment Plants" [28] and international standards [29] and it was concluded that there is no harm in the use of these waters for irrigation purposes (Table 2).

## Experimental design and irrigation practices

The study was established according to a completely randomized design with three replications and five different irrigation water treatments were applied (Table 3). The treatments were obtained by mixing treated wastewater (WW) and clean tap water (CW) in different proportions and were planned as follows: 100% WW (treated wastewater only), 75% WW + 25% CW, 50% WW + 50% CW, 25% WW + 75% CW and 100% CW (clean water only-control). The soil used in the experiment was shade-dried, then prepared by sieving through a 2 mm diameter sieve and placed in plastic pots of 15 liters each. The pots were kept under outdoor conditions and watering was carried out according to the specified rates. A pot-based experimental model was selected in this study to ensure precise control over the irrigation regimes, particularly the ratio of treated wastewater to clean water applied over a multi-year period. This approach enabled consistent treatment application, minimized external environmental variability, and allowed systematic sampling of soil for heavy metal analysis over four consecutive years. However, we acknowledge that pot experiments may not fully simulate natural field-scale

**Table 2. Irrigation water quality parameters.**

| Property | W1 | W2 | W3 | W4 | W5 |
|---|---|---|---|---|---|
| pH | 7.48 | 7.51 | 7.52 | 7.55 | 7.65 |
| EC (ds m$^{-1}$) | 0.68 | 0.61 | 0.56 | 0.52 | 0.49 |
| SAR | 2.01 | 1.72 | 1.44 | 0.83 | 0.45 |
| Ca (meq L$^{-1}$) | 2.88 | 2.25 | 1.94 | 1.34 | 0.96 |
| Mg (meq L$^{-1}$) | 1.14 | 0.83 | 0.68 | 0.41 | 0.32 |
| Na (meq L$^{-1}$) | 2.85 | 2.14 | 1.65 | 0.78 | 0.36 |
| K (meq L$^{-1}$) | 0.27 | 0.22 | 0.2 | 0.19 | 0.17 |
| $CO_3$ (meq L$^{-1}$) | – | – | – | – | – |
| $HCO_3$ (meq L$^{-1}$) | 4.02 | 1.85 | 0.42 | 0.33 | 0.24 |
| Cl (meq L$^{-1}$) | 2.02 | 1.63 | 1.02 | 0.66 | 0.51 |
| $SO_4$ (meq L$^{-1}$) | 0.28 | 0.16 | 0.14 | 0.12 | 0.12 |
| B (mg L$^{-1}$) | 0.28 | 0.13 | 0.012 | 0.07 | 0.0001 |
| Fe (mg L$^{-1}$) | 0.064 | 0.041 | 0.033 | 0.0084 | 0.0035 |
| Cu (mg L$^{-1}$) | 0.003 | 0.002 | 0.002 | – | – |
| Mn (mg L$^{-1}$) | 0.006 | 0.003 | 0.003 | 0.001 | 0.001 |
| Zn (mg L$^{-1}$) | 0.002 | 0.001 | 0.001 | – | – |
| Pb (mg L$^{-1}$) | 0.0013 | 0.001 | – | – | – |
| Cd (mg L$^{-1}$) | – | – | – | – | – |
| Cr (mg L$^{-1}$) | 0.001 | 0.001 | – | – | – |
| Ni (mg L$^{-1}$) | 0.003 | 0.0024 | 0.001 | – | – |
| COD (mg L$^{-1}$) | 32.61 | 15.89 | 8.24 | 3.78 | – |
| $BOD_5$ (mg L$^{-1}$) | 13.82 | 6.75 | 3.29 | 1.23 | – |

WW: treated wastewater, CW: clean water,-: not detectd, EC: electrical conductivity, SAR: sodium adsorption ratio, COD: chemical oxygen demand, $BOD_5$: biological oxygen demand, W1: 100% WW, W2: 75% WW+25% CW, W3: 50% WW+50% CW, W4: 25% WW+75% CW, W5: 100% CW

**Table 3. Concentrations of irrigation treatment.**

| No. Treatments | Concentrations |
|---|---|
| W1 | 100% WW |
| W2 | 75% WW+25% CW |
| W3 | 50% WW+50% CW |
| W4 | 25% WW+75% CW |
| W5 | 100% CW |

WW: treated wastewater, CW: clean water.

hydrological processes, such as deep percolation, runoff, and soil heterogeneity. Therefore, the results should be interpreted with consideration of these limitations, and further field-based research is recommended to validate the observed heavy metal accumulation patterns under real agricultural conditions.

The experimental investigation commenced in 2021; soil samples were collected and evaluated subsequent to the applications. The identical irrigation applications were replicated in 2022, 2023, and 2024 to assess the cumulative effect and were consistently maintained for a total duration of four years. The applications were consistently organized annually, and soil studies were conducted at the conclusion of each season. Within the scope of the experiment, only one season of irrigation was carried out in 2021; the same irrigation pattern was maintained for two seasons between 2021 and 2022, three seasons between 2021 and

2023, and four seasons between 2021 and 2024. Thus, the change in heavy metal accumulation over the years was evaluated in soils irrigated with treated wastewater for different periods. During the experiment, soil moisture in the pots was maintained at a level close to field capacity, and periodic irrigation was carried out according to the plants' water requirements.

This four-year field experiment employs a long-term method to evaluate the cumulative impact of heavy metal deposition in soils subjected to wastewater over varying durations. The treatments were administered for one season in 2021, two seasons in 2022, three seasons in 2023, and four seasons in 2024, facilitating a comparative analysis of accumulation across the years.

### Soil analysis

At the end of each irrigation season, soil samples were collected from three different points within each replicate pot. These three subsamples were thoroughly homogenized to form one composite sample per pot. As a result, three composite samples were obtained for each irrigation treatment per year. In total, 15 composite soil samples were prepared and analyzed annually. This procedure was repeated over four consecutive years, resulting in a total of 60 composite soil samples analyzed throughout the study period. The collected samples were air-dried, passed through a 2 mm mesh sieve, and prepared for heavy metal analysis. The concentrations of Cr, Ni, Cu, Zn, Cd, Fe, and Pb in the soil were determined using ICP-OES, following the method described by [30].

### Assessment indices for soil heavy metal contamination

In order to assess heavy metal contamination in soils, various individual indices are used, such as the geoaccumulation index (Igeo), enrichment factor (EF), pollution load index (PLI) and contamination factor (CF). These indices are calculated based on the concentration of each metal present in the soil and are used to classify pollution levels (Table 4) [31–34].

where CF is contamination factor, $C_i$ is metal concentration at a contaminated site; $C_{i\ (background)}$ is concentration of a given element in background sample. EF value is used to differentiate magnitude of contamination resulting from either the natural or human influence. where $(Ci/CFe)_{(sample)}$ is the ratio of the concentration of a metal with that of Fe at each sampling point, $(Ci/CFe)_{(background)}$ is the same ratio of the concentrations in the reference environment. In EF calculations, commonly used reference metals include Mn, Al, and Fe. In this study, iron (Fe) was selected as the reference element, following the approach of [35]. Igeo includes a constant factor of 1.5, which compensates for background variations stemming from natural lithogenic sources. PLI is the pollution load index, where CF: contamination factor, n: number of study metals.

The natural background concentration $C_{i\ (background)}$ used to assess soil pollution is an important reference measure. In some studies, this value has been determined based on global averages [14,36]. The values used in the present study were based on global average background concentrations as reference values. However, in regions with different geological and soil structures, researchers also use local background values to obtain more accurate results [37,38]. This method allows pollution to be assessed on a region-specific basis.

### Statistical analysis

The data obtained as a result of the research were subjected to analysis of variance with the help of SPSS computer program and compared with analysis of variance (ANOVA) and Duncan multiple comparison test for differences between means. The results of the traits were recorded as statistically significant at 5% significance level. Statistical tests were performed using SPSS version 22.0 statistical program.

## Results and discussion

### Levels of heavy metals in soil

In this study, heavy metal contents were determined in soil samples taken from pots at the end of the 2021, 2022, 2023, and 2024 seasons following irrigation treatments applied according to different water quality concentrations. The results

**Table 4. Classes and equations of single indices: CF, EF, I$_{geo}$, PLI.**

| Indices | Equations | Limit values | Soil quality |
|---|---|---|---|
| CF | $CF = \dfrac{C_i}{C_{i\ (background)}}$ | CF < 1 | Low contamination factor |
| | | $1 \leq CF \leq 3$ | Moderate contamination factor |
| | | $3 \leq CF \leq 6$ | Considerable contamination factor |
| | | $6 \leq CF$ | Very high contamination factor |
| EF | $EF = \dfrac{\left(\frac{C_i}{C_{Fe}}\right)(sample)}{\left(\frac{C_i}{C_{Fe}}\right)(background)}$ | EF < 2 | Deficiency to minimal mineral enrichment |
| | | 2-5 | Moderate enrichment |
| | | 5-20 | Significant enrichment |
| | | 20-40 | Very high enrichment |
| | | EF > 40 | Extremely high enrichment |
| I$_{geo}$ | $I_{geo} = log_2\left(\dfrac{C_i}{1.5\ C_{i\ (background)}}\right)$ | Igeo ≤ 0 | Uncontaminated |
| | | 0 ≤ Igeo < 1 | Uncontaminated to moderately Contaminated |
| | | 1 ≤ Igeo < 2 | Moderately contaminated |
| | | 2 ≤ Igeo < 3 | Moderately to strongly contaminated |
| | | 3 ≤ Igeo < 4 | Strongly contaminated |
| | | 4 ≤ Igeo < 5 | Strongly to extremely Contaminated |
| | | Igeo > 5 | Extremely high contaminated |
| PLI | $PLI = \sqrt[n]{CF_1\ x\ CF_2\ x\ldots x\ CF_n}$ | 0 < PLI ≤ 1 | Unpolluted |
| | | 1 < PLI ≤ 2 | Unpolluted to moderate |
| | | 2 < PLI ≤ 3 | Moderate polluted |
| | | 3 < PLI ≤ 4 | Moderate to highly polluted. |

CF: contamination factor, EF: enrichment factor, I$_{geo}$: geographic accumulation index, PLI: pollution load index.

obtained are presented in Table 6. The limit values for heavy metal concentrations specified by the Republic of Türkiye Ministry of Environment [28,39] and Urbanization and Climate Change and Urbanization and European Union Standards [40,41] are given in Table 5.

Statistically significant differences were observed in the accumulation levels of Cr, Ni, Cu, Zn, Cd, Fe and Pb in the soil as a result of irrigation water mixtures with different ratios applied for four years. There was no statistically signifi-cant difference between heavy metal concentrations measured in 100% CW irrigated soils for four years. According to Table 6, the concentrations of heavy metals determined in soil samples vary according to the elements. Cr concentration ranged between 4.08–59.41 mg kg$^{-1}$ with an average value of 20.24 mg kg$^{-1}$. Ni ranged between 0.79–8.59 mg kg$^{-1}$ with an average of 4.34 mg kg$^{-1}$. Cu was between 2.98–35.66 mg kg$^{-1}$ (mean 14.83 mg kg$^{-1}$) and Zn between 55.14–109.66 mg kg$^{-1}$ (mean 77.69 mg kg$^{-1}$ mg kg$^{-1}$). Cd concentrations ranged between 0.00–1.25 mg kg$^{-1}$, with an average of 0.56 mg kg$^{-1}$. Fe showed the highest value among all elements, ranging from 21,995.00–80,339.00 mg kg$^{-1}$, with an average of 44,533.61 mg kg$^{-1}$. Pb was found in the range of 0.01–55.88 mg kg$^{-1}$ with an average value of 14.64 mg kg$^{-1}$. According to Table 5, the heavy metal concentrations determined in the study were found to be below the maximum permissible limit values specified in the compared national and international standards.

The mobility and retention of heavy metals in wastewater-irrigated soils are primarily controlled by soil pH, electrical conductivity (EC), cation exchange capacity (CEC), and organic matter (OM). In semi-arid regions, low OM and increas-ing salinity due to repeated irrigation amplify these effects [42,43]. Lower pH enhances metal solubility and mobility, while higher pH and elevated CEC promote adsorption through precipitation and surface complexation, especially in clay-rich

**Table 5. Permissible heavy metal limit values in soil.**

| Standard | | Cd (mg kg⁻¹) | Cr (mg kg⁻¹) | Cu (mg kg⁻¹) | Pb (mg kg⁻¹) | Ni (mg kg⁻¹) | Zn (mg kg⁻¹) |
|---|---|---|---|---|---|---|---|
| EU 2002 | | 3 | 150 | 140 | 300 | 75 | 300 |
| Türkiye | $6 \leq pH < 7$ | 1 | 60 | 50 | 70 | 50 | 150 |
| | $pH \geq 7$ | 1.5 | 100 | 100 | 100 | 70 | 200 |

**Table 6. Total heavy metal content (mg kg⁻¹) of potting soils.**

| Years | Water Quality | Heavy metal concentrations (mg kg⁻¹) | | | | | | |
|---|---|---|---|---|---|---|---|---|
| | | Cr | Ni | Cu | Zn | Cd | Fe | Pb |
| 1Y | W1 | 34.67±0.22 aD | 4.31±0.04 aD | 21.73±0.06 aD | 97.62±1.13 aD | 0.85±0.02 aD | 53,776.00±325 aD | 24.12±0.34 aD |
| | W2 | 19.36±0.34 bD | 4.03±0.07 bD | 18.65±0.492 bD | 80.22±0.51 bD | 0.48±0.04 bD | 49,775.00±458 bD | 16.41±0.20 bD |
| | W3 | 13.53±0.30 cD | 2.96±0.04 cD | 10.06±0.08 cD | 62.69±0.07 cD | 0.39±0.02 cD | 35,483.00±141 cD | 2.09±0.05 cD |
| | W4 | 7.05±0.03 dD | 2.30±0.03 dD | 7.77±0.09 dD | 59.44±0.21 dD | 0.11±0.02 dD | 23,344.00±142 dD | 0.84±0.02 dD |
| | W5 | 4.08±0.04 eNs | 0.79±0.02 eNs | 2.98±0.07 eNs | 55.14±0.07 eNs | 0.00±0.00 eNs | 21,995.00±75.2 eNs | 0.01±0.00 eNs |
| 2Y | W1 | 45.19±0.45 aC | 5.58±0.16 aC | 26.46±0.263 aC | 103.37±0.29 aC | 1.03±0.05 aC | 58,545.90±279 aC | 36.23±0.35 aC |
| | W2 | 21.55±0.25 bC | 5.08±0.11 bC | 19.14±0.172 bC | 85.30±0.66 bC | 0.64±0.04 bC | 50,216.00±167 bC | 18.95±0.34 bC |
| | W3 | 16.41±0.11 cC | 4.12±0.04 cC | 11.14±0.02 cC | 71.99±0.13 cC | 0.55±0.02 cC | 36,932.00±153 cC | 3.02±0.03 cC |
| | W4 | 7.23±0.55 dC | 3.11±0.02 dC | 8.23±0.05 dC | 63.87±0.27 dC | 0.14±0.01 dC | 26,741.00±386 dC | 0.92±0.07 dC |
| | W5 | 4.21±0.05 eNs | 0.86±0.02 eNs | 3.05±0.05 eNs | 55.65±0.07 eNs | 0.00±0.00 eNs | 22,054.00±85.8 eNs | 0.01±0.00 eNs |
| 3Y | W1 | 51.23±0.30 aB | 6.98±0.55 aB | 33.57±0.37 aB | 106.68±0.57aB | 1.18±0.04 aB | 70,616.00±275 aB | 48.33±0.22 aB |
| | W2 | 25.99±0.11 bB | 6.51±0.06 bB | 20.53±0.05 bB | 92.13±0.58 bB | 0.77±0.05 bB | 58,706.00±90.6 bB | 28.99±0.51 bB |
| | W3 | 18.85±0.21 cB | 5.49±0.03 cB | 13.28±0.14 cB | 76.48±0.20 cB | 0.60±0.02 cB | 44,693.00±63.9 cB | 5.04±0.07 cB |
| | W4 | 8.69±0.07 dB | 4.19±0.08 dB | 9.69±0.08 dB | 66.66±2.90 dB | 0.24±0.05 dB | 31,472.00±230 dB | 2.11±0.02 dB |
| | W5 | 4.25±0.04 eNs | 0.82±0.02 eNs | 3.03±0.02 eNs | 56.04±0.35 eNs | 0.00±0.00 eNs | 22,244.00±101 eNs | 0.01±0.00 eNs |
| 4Y | W1 | 59.41±0.19 aA | 8.59±0.04 aA | 35.66±0.04 aA | 109.66±0.41 aA | 1.25±0.04 aA | 80,339.00±96.3 aA | 55.88±0.63 aA |
| | W2 | 28.41±0.12 bA | 7.54±0.02 bA | 21.77±0.02 bA | 98.18±0.04 bA | 0.89±0.02 bA | 72,672.07±61.4 bA | 37.24±0.32 bA |
| | W3 | 21.33±0.03 cA | 6.92±0.02 cA | 15.77±0.02 cA | 81.36±0.13 cA | 0.66±0.03 cA | 64,504.32±84.5 cA | 8.36±0.04 cA |
| | W4 | 9.26±0.26 dA | 5.87±0.03 dA | 11.11±0.03 dA | 75.32±0.06 dA | 0.28±0.01 dA | 44,419.00±85.2 dA | 4.25±0.02 dA |
| | W5 | 4.18±0.03 eNs | 0.84±0.02 eNs | 3.06±0.02 eNs | 56.10±0.26 eNs | 0.00±0.00 eNs | 22,145.00±110 eNs | 0.01±0.00 eNs |
| P value | Years (Y) | *** | *** | *** | *** | *** | *** | *** |
| | Water Quality (WQ) | *** | *** | *** | *** | *** | *** | *** |
| | Y * Q | *** | *** | *** | *** | *** | *** | *** |

1Y: In 2021, only one season of irrigation was applied, 2Y: Irrigation was practiced for two seasons between 2021 and 2022, 3Y: irrigated for three seasons between 2021–2023, 4Y: Irrigation for four seasons in 2021–2024, WW: treated wastewater, CW: clean water, W1: %100WW, W2: %75WW+%25CW, W3: %50WW+%50CW, W4: %25WW+%75CW, W5: %100CW. Different lower case letters in the same column indicate that there is a significant difference between the means of different water quality treatments and upper case letters indicate that there is a significant difference between the means of different years at p<0.05 level by Duncan test (Mean±SD). Ns: not significant

soils [44,45]. The observed variation in pollution indices across treatments may partly reflect these pH-dependent sorption processes. Elevated EC values, typical of wastewater-irrigated soils, increase ion competition and metal desorption, whereas reduced OM limits the formation of stable metal–organic complexes that would otherwise immobilize metals [46,47]. Similar patterns were reported by [35], who found higher exchangeable cations and ESP under full wastewater irrigation. The positive relationship between OM and CEC [48,49] and the rise in ESP under saline conditions support our findings and agree with studies from Mediterranean and semi-arid regions emphasizing risks of sodicity and reduced soil productivity [38,50].

Heavy metal contents gradually increased both as the year of application progressed and as the proportion of treated wastewater increased.

In this study, it was determined that heavy metal contents in the soil increased both as the year of application progressed and as the rate of treated wastewater used increased. Similarly, it is stated in the literature that irrigation with wastewater increases heavy metal accumulation in soil [22,51]. This is an expected result since wastewater can contain metals from various sources. Previous studies have shown that heavy metal levels in soils irrigated with wastewater were higher than those irrigated with clean water. For example, [52] reported that long-term irrigation with treated wastewater in Mediterranean conditions significantly altered soil physicochemical and microbiological properties, with total Ni, Zn, Cu, Pb, and Cd concentrations increasing proportionally with irrigation years (though generally remaining below Tunisian standards). They concluded that the magnitude of these changes was closely related to the duration of wastewater application, highlighting the necessity of proper management and periodic monitoring to ensure the safe and sustainable reuse of wastewater in agriculture. Similarly, [53] demonstrated that chemical analyses revealed significant accumulation of heavy metals such as Cd, Cu, Zn, and Ni in soils irrigated with wastewater for 20 years, indicating that trace element buildup could pose a serious threat to soil biological processes and may act as a limiting factor for the long-term reuse of wastewater in agriculture. These findings further emphasize the importance of continuous monitoring and careful management when integrating recycled wastewater into irrigation practices. These findings are in agreement with our study [54,55].

During all seasons, the highest heavy metal values were detected in pot soils irrigated with W1, while the lowest concentrations were recorded in the control group irrigated with W5. Accumulation levels were more limited especially in W3 and W4 treatments, indicating that reducing the mixing ratio can control accumulation. Throughout the experiment, it was determined that heavy metal concentrations measured in the soil on a yearly basis increased compared to the previous year. This indicates that the application of treated wastewater in consecutive years in the same plots led to cumulative heavy metal accumulation in the soil. In the study, it was observed that although some heavy metals (especially Fe and Pb) tended to accumulate in irrigation with treated wastewater, the concentrations of elements such as Cr, Ni, Cu, Zn, Cd and Pb remained largely below the accepted limit values.

Many studies have reported that the heavy metal contents of soils irrigated with wastewater are below the permissible limits of international regulations [24,26,56]. In the literature, different findings on heavy metal accumulation in soils irrigated with wastewater are also found. For example, [25] reported that metals such as Zn, Cu, Pb and Cd were found above permissible limits in soils irrigated with treated wastewater and that soils were contaminated with these metals. Similarly, [57] reported increased accumulation of heavy metals in soils in areas where wastewater has been used continuously for more than 20 years, resulting in Cd, Pb and Ni concentrations exceeding the limits considered safe for human health in grown vegetables. However, some studies present more moderate results. [12] reported that heavy metal concentrations in soils irrigated with treated wastewater were below limits in some areas and above in others. The researchers emphasized that in the short term, these accumulations do not pose a serious risk to soil and crops; however, caution should be exercised about possible effects on human health in long-term applications. It is also reported that heavy metal accumulation in soils irrigated with wastewater may vary depending on the physicochemical properties of the soil and the duration of irrigation [58]. Therefore, irrigation with wastewater should be considered not only for its impact on soil pollution but also for its potential effects on crop safety and human health through the accumulation of contaminants in plant tissues. The bioaccumulation of heavy metals and pollutants in plants due to wastewater irrigation represents a significant environmental and public health concern. As urban areas continue to expand, the reuse of treated and untreated wastewater for agricultural purposes has become prevalent. However, this practice poses risks of heavy metal accumulation in crops, which can subsequently impact human health upon consumption [57]. Prolonged irrigation with wastewater has been shown to cause the accumulation of heavy metals such as lead (Pb), cadmium (Cd), copper (Cu), and zinc (Zn) in soils, which can subsequently accumulate in plants [59,60]. Such accumulation may be transmitted into the human food chain by consumption of the products and may constitute a health hazard. Indeed, [61] reported high levels of heavy

metals in vegetables irrigated with sewage, which indicate a serious risk for public health. Similarly, [62] found a correlation between the level of heavy metals in the soil of alfalfa plants and the growth of plant tissue. It was also stressed that although nutrient-rich waste water (N, P, K) contributes to increased yields, excessive accumulation of waste water can have negative consequences for soil quality and food safety [63–65].

In this context, studies have shown that well-managed treated sewage can have a positive impact on nutrient uptake and yields and reduce pathogens and heavy metals [66–68]. [69] reported that treated sewage increased the total enzymatic activity of nitrogen, phosphorus and soil enzymes in tomato plants. Similarly, [70] reported that municipal waste water improves the soil characteristics and yields of chickpea, wheat and rye crops. However, the mandatory use of untreated sewage, particularly in developing countries, continues to pose a serious health risk for consumers [71–73]. Controlled use of treated sewage is therefore essential for sustainable and safe practices, as are long-term monitoring, awareness-raising of farmers and assessment of food safety parameters [74]. Moreover, dietary exposure to heavy metals from contaminated crops raises significant health concerns. Long-term consumption can lead to various health issues, ranging from acute toxicity to chronic diseases such as kidney damage, neurological impairments, and potential carcinogenic effects [65,75]. A health risk assessment surrounding the use of wastewater in agriculture suggests that even low concentrations of heavy metals can accumulate to dangerous levels within the food chain, potentially affecting both human and livestock health [76]. In general, while the practice of using wastewater for irrigation may offer advantages in terms of nutrient supply and water conservation, it carries notable risks of heavy metal bioaccumulation in crops that threaten public health. Continuous monitoring and stringent regulatory practices are essential to ensure that agricultural activities using wastewater do not compromise food safety and human health.

## Soil heavy metal pollution indices

CF, EF, Igeo, and PLI indices were calculated to evaluate heavy metal accumulation in the soil depending on different irrigation water quality concentrations and years; the findings are presented in Figs 3-6, respectively.

Heavy metal accumulation in soil was evaluated according to the CF index (Fig 3), depending on different irrigation water quality and application period. In general, CF values increased with increasing treated wastewater (WW) rate and year of application. The highest CF values were observed in the W1 treatment group; especially Pb, Fe, Zn and Cd elements approached the intermediate pollution limits at the end of the fourth year. The CF values of Cr, Ni and Cu were at low risk of contamination in all treatments. In the control groups irrigated with W5, the CF values of all heavy metals remained at very low levels. In the treatments with different concentrations (W2 and W3), some metals showed values close to the limit but mostly below 1.00. This indicates that controlled use of treated wastewater can significantly limit soil pollution.

EF (Enrichment Factor) analyses performed to evaluate heavy metal enrichment in soil showed variability according to different irrigation water quality and application period (Fig 4). When EF values were analyzed, minimal enrichment was observed for Cr, Ni, Cu, Zn, Cd and Pb in all treatments. EF values showed an increasing trend especially in the treatments with high wastewater (WW) ratio and as the application period increased. In mixed irrigation treatments (W2 and W3), EF values were generally below 1.00, especially for Cr, Ni and Cu these values were found below 0.5. No significant enrichment was observed even for potentially risky metals such as Pb and Zn. This indicates that these metals are largely of natural origin. In the W5 treatment, EF values for all metals were below 1.00, indicating that there was no exogenous enrichment in the soil.

Igeo, which is used to evaluate heavy metal pollution in soil in comparison with past values, was calculated according to different irrigation water qualities and application periods (Fig 5). The Igeo values obtained for all elements were in the negative range, well below zero. The Igeo values of all metals in the control groups irrigated with W5 were found to be the lowest without exception. Especially for Pb, the Igeo value was well below the background value in all years, indicating no pollutant effect whatsoever. In the W1 treatments, Igeo values were relatively higher; a gradual increase was observed

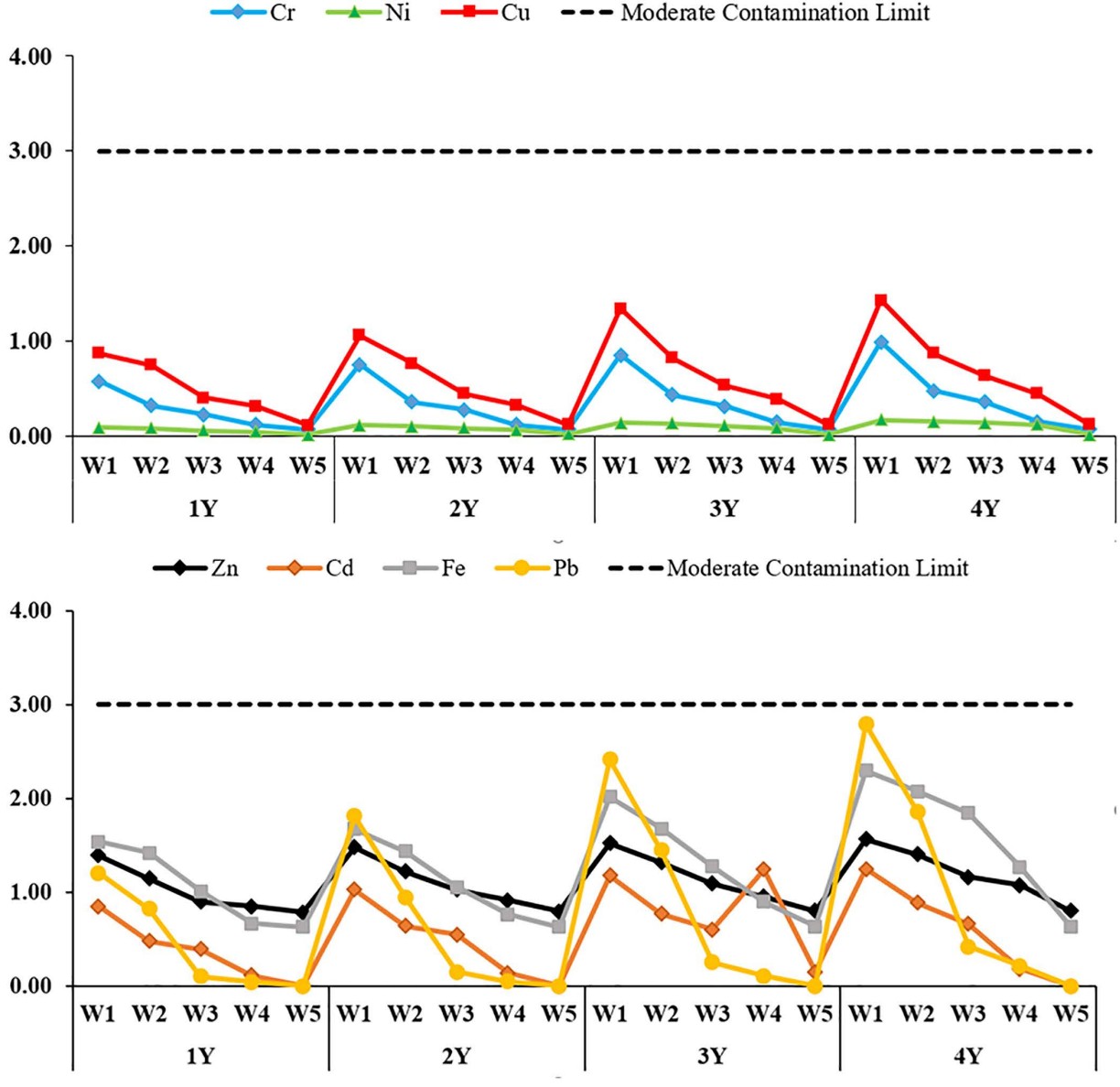

**Fig 3. Contamination factor (CF) values for soil heavy metals for different water qualities and years.** 1Y: In 2021, only one season of irrigation was applied, 2Y: Irrigation was practiced for two seasons between 2021 and 2022, 3Y: irrigated for three seasons between 2021-2023, 4Y: Irrigation for four seasons in 2021-2024, WW: treated wastewater, CW: clean water, W1: %100WW, W2: %75WW+%25CW, W3: %50WW+%50CW, W4: %25WW+%75CW, W5: %100CW.

for Cu, Pb and Cr for four years. However, despite this increase, all Igeo values remained below 0 and were classified as "unpolluted". This indicates that although four years of continuous wastewater application has caused a visible accumulation of metals in the soil, this accumulation has not yet constituted geochemically significant pollution. In mixed rate irrigation treatments (W2 and W3), Igeo values were generally lower than in the W1 group, but higher than in the W5 group. Although slight increases in metals such as Cr, Ni and Cu were observed especially in the 2nd and 3rd years, these increases did not reach significant pollution levels (Igeo<0).

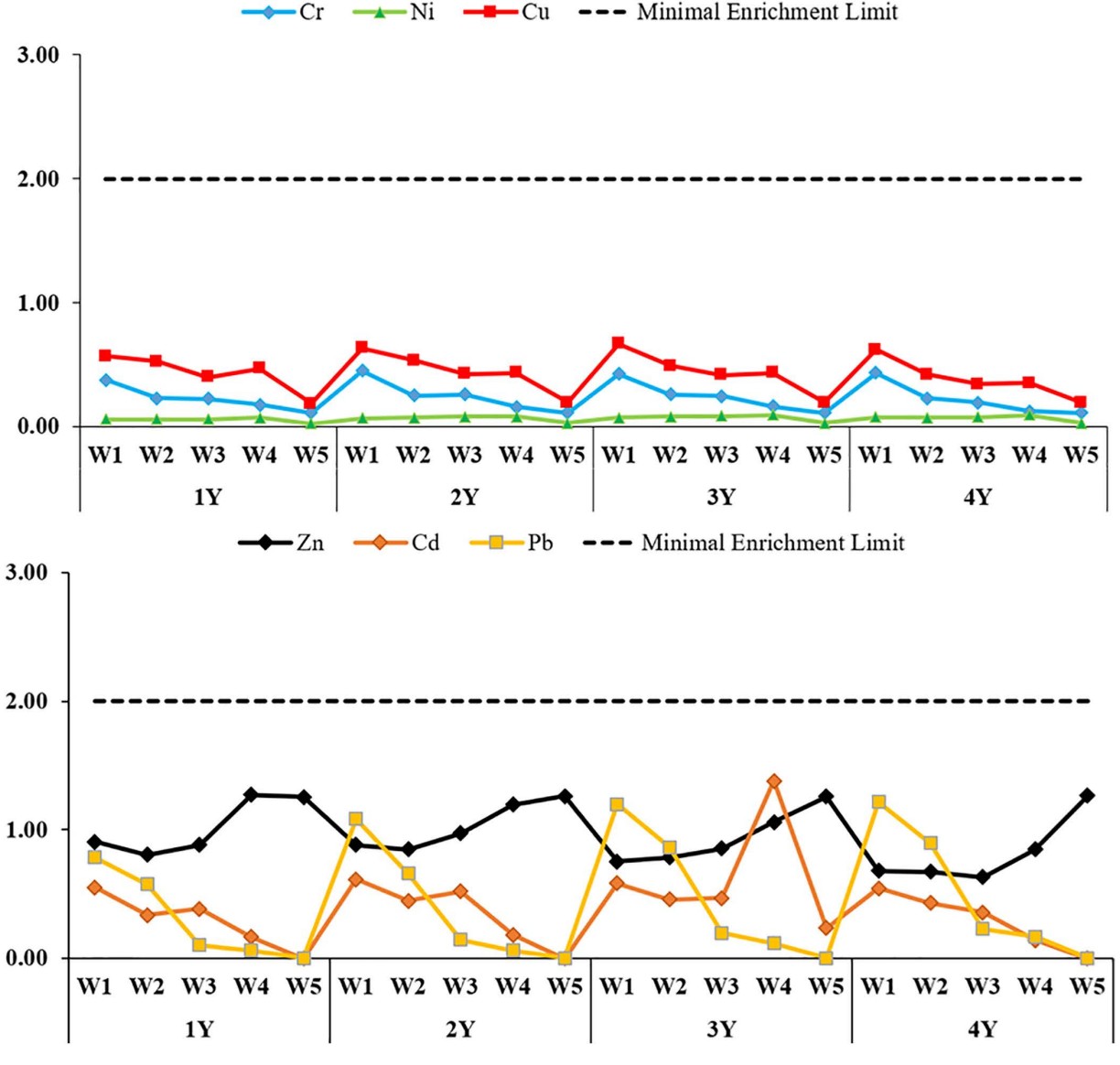

**Fig 4. Enrichment factor (EF) values for soil heavy metals for different water qualities and years.** 1Y: In 2021, only one season of irrigation was applied, 2Y: Irrigation was practiced for two seasons between 2021 and 2022, 3Y: irrigated for three seasons between 2021-2023, 4Y: Irrigation for four seasons in 2021-2024, WW: treated wastewater, CW: clean water, W1: %100WW, W2: %75WW+%25CW, W3: %50WW+%50CW, W4: %25WW+%75CW, W5: %100CW.

Pollution Load Index (PLI) values calculated to assess the overall contamination level in soils are presented in Fig 6 for different irrigation water mixtures and years of application. In general, PLI values increased as the wastewater (WW) rate increased and as the application period increased. The highest PLI values were observed in the treatments irrigated with 100% WW for each year, and the PLI value approached the limit of moderate contamination at the end of the fourth year. In the control group irrigated with W5, PLI < 1.00 was maintained in all years, indicating that there was no accumulation of heavy metal contamination in the soils. The PLI values calculated in all treatments remained below 1.00, indicating that the soils were generally classified as unpolluted. In different studies conducted in similar regions, it has been reported that soil heavy metal pollution indexes are mostly at low levels and do not pose a serious environmental risk [35,38]. These findings

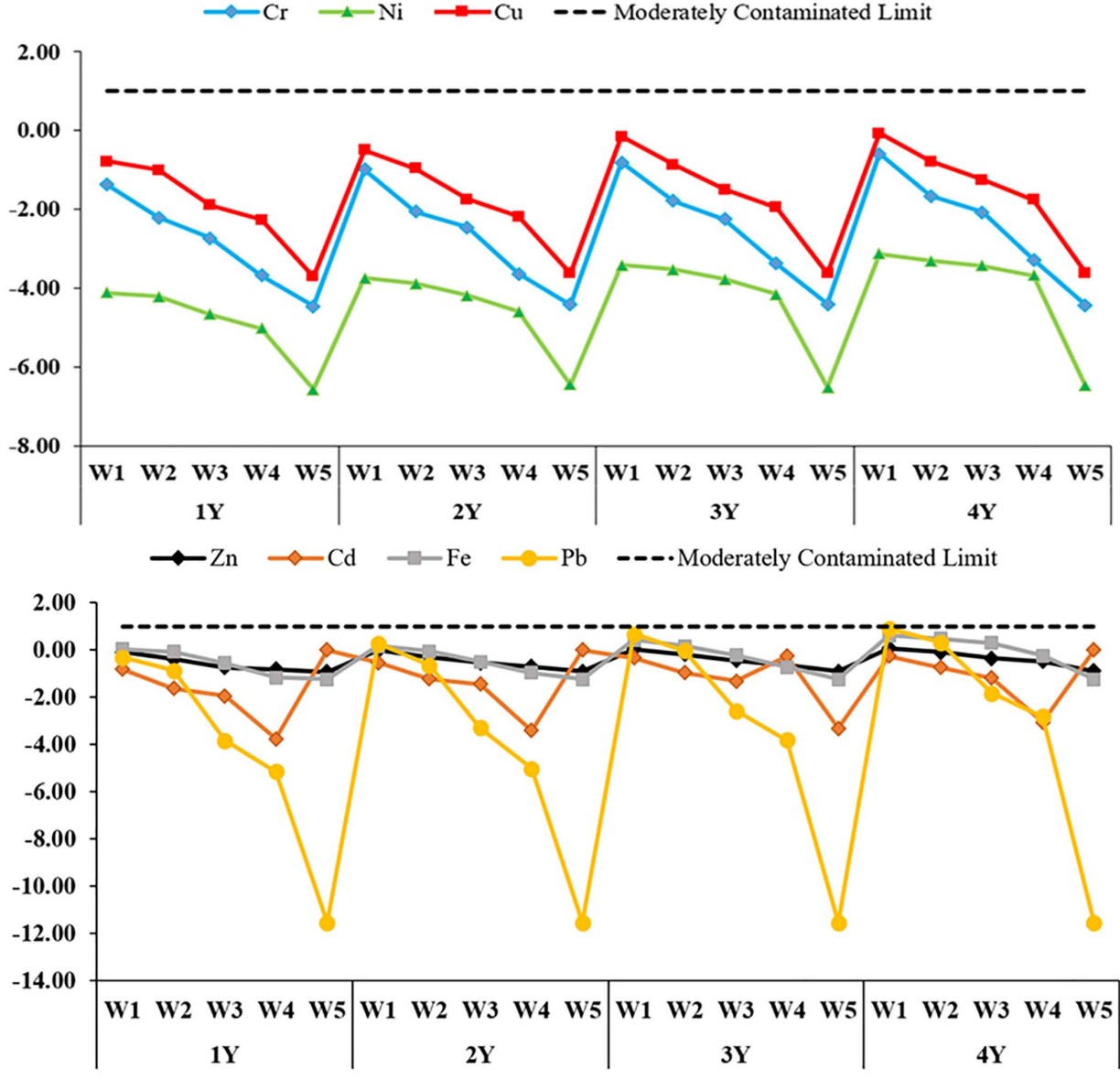

**Fig 5. Geoaccumulation index ($I_{geo}$) values for soil heavy metals for different water qualities and years.** 1Y: In 2021, only one season of irrigation was applied, 2Y: Irrigation was practiced for two seasons between 2021 and 2022, 3Y: irrigated for three seasons between 2021-2023, 4Y: Irrigation for four seasons in 2021-2024, WW: treated wastewater, CW: clean water, W1: %100WW, W2: %75WW + %25CW, W3: %50WW + %50CW, W4: %25WW + %75CW, W5: %100CW.

are consistent with the results obtained in our study. Similar findings were also found in studies conducted in different regions. For example, [16] reported that the soils of the Aswan-Luxor region were classified as moderately contaminated according to the Igeo index; CF values indicated low contamination levels for As, Cu and Ni, medium for Fe and Co, and high for Pb, Cr, Cd and Zn. EF values were reported to be at low enrichment level, while the PLI index indicated that the region was generally moderately contaminated. Similarly, [37] reported that in La Zanja region, 67.7% of the soils were unpolluted, 17.8% were slightly polluted and only 5.45% were moderately polluted for Cr, As and Pb. [34] reported that only 19% of the soil samples were moderately contaminated for Cd and Cr, while other metals were generally classified as unpolluted or

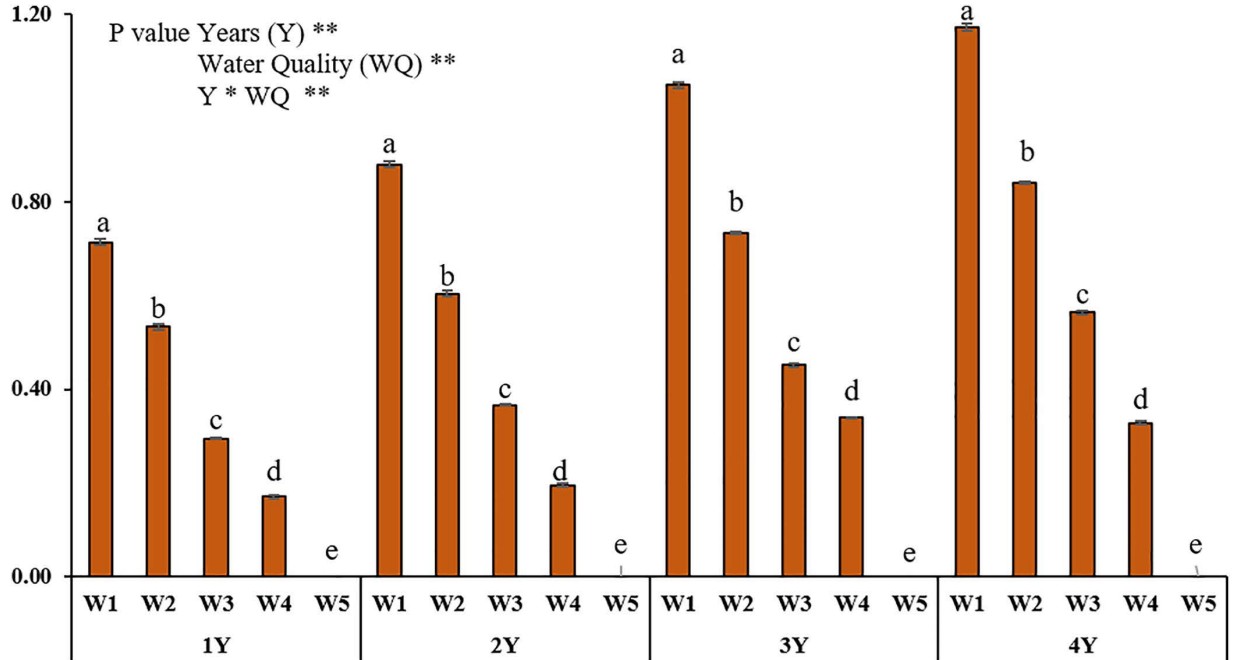

**Fig 6. Pollution Load Index (PLI) values for soil heavy metals for different water qualities and years.** 1Y: In 2021, only one season of irrigation was applied, 2Y: Irrigation was practiced for two seasons between 2021 and 2022, 3Y: irrigated for three seasons between 2021-2023, 4Y: Irrigation for four seasons in 2021-2024, WW: treated wastewater, CW: clean water, W1: %100WW, W2: %75WW + %25CW, W3: %50WW + %50CW, W4: %25WW + %75CW, W5: %100CW. Different letters indicate significant differences (p < 0.05).

slightly contaminated. In addition, PLI values below 1 in surface soils revealed that these areas were generally unpolluted. Although this study focused solely on the accumulation of heavy metals in soil, it did not include an analysis of metal uptake by plants. Future research should investigate the potential bioaccumulation of these metals in plant tissues to better assess the food safety and ecological risks associated with the long-term use of treated wastewater in agricultural irrigation.

This study was conducted using a pot-based experimental setup, which offered a high level of control over irrigation regimes—particularly the treated wastewater and clean water ratios—and allowed for systematic and repeated soil sampling over four consecutive years. However, it is important to emphasize that this approach does not fully replicate field-scale hydrological and biological processes, such as deep percolation, surface runoff, soil heterogeneity, and complex root-soil interactions that naturally occur under open field conditions. These factors can significantly influence the mobility, retention, and bioavailability of heavy metals in soil. Additionally, the study focused solely on the accumulation of heavy metals in the soil compartment, without analyzing their uptake and translocation in plant tissues. This represents a limitation in fully assessing the potential human and ecological risks related to bioaccumulation through the food chain. Therefore, while the findings offer valuable insights under controlled conditions, caution must be exercised when extrapolating these results to real-world agricultural systems. Future research should involve long-term, field-based trials that incorporate both soil and plant analyses to more comprehensively evaluate the environmental sustainability, crop safety, and potential health implications of treated wastewater reuse in agriculture.

## Conclusion

This study comprehensively evaluated the effects of different proportions of treated wastewater and clean water mixtures on heavy metals accumulated in soil as a result of 4-year field experiment irrigation practices. The findings revealed that the accumulation levels of heavy metals (especially Cd, Ni, Pb and Zn) increased significantly in 100% WW treatments

and some metals approached the medium contamination category. In contrast, the mixed rate irrigation treatments (W2 and W3) significantly reduced the heavy metal accumulation levels in the soil and were classified as low risk according to the CF, EF, Igeo and PLI indices, while the W5 treatments showed non-contaminated or very low contamination levels in all indices. Based on the applied pollution indices (CF, EF, Igeo, and PLI), irrigation with 25–50% treated wastewater mixtures can be considered environmentally safe over a four-year period, as heavy metal accumulation remained below national and international contamination thresholds. Therefore, farmers and agricultural planners in water-scarce regions may adopt these blending ratios as a sustainable irrigation strategy. However, continued use beyond four years may lead to progressive accumulation of certain metals (e.g., Cd and Pb), particularly under higher WW proportions. To ensure long-term soil health and prevent possible food chain contamination, it is recommended to implement periodic soil monitoring, rotate irrigation sources, and integrate phytoremediation or crop rotation techniques in future practices. Further long-term studies under field conditions are also encouraged to validate the scalability of these findings.

The findings suggest that the direct application of wastewater may jeopardize soil quality and environmental sustainability over time; yet, these concerns can be mitigated when combined with clean water in regulated ratios. In this context, it is advisable to examine diverse irrigation practices as a significant alternative for the sustainable utilization of water resources and the preservation of soil health in agricultural production. Although the findings of this study provide valuable insights into the dynamics of heavy metal accumulation under controlled irrigation with treated wastewater, it must be emphasized that the pot-based experimental design does not fully replicate field-scale hydrological and biological conditions. Therefore, the conclusions drawn here should be interpreted with caution, and future research should focus on long-term field experiments to validate these results under real agricultural conditions.

## Author contributions

**Conceptualization:** Hasan Er.

**Data curation:** Hasan Er.

**Investigation:** Hasan Er.

**Methodology:** Hasan Er.

**Writing – original draft:** Hasan Er.

**Writing – review & editing:** Hasan Er.

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
