## [Decision Letter · Decision Letter 0]

15 Sep 2025

Dear Dr. Er,

Thank you for submitting your manuscript to PLOS ONE. After careful consideration, we feel that it has merit but does not fully meet PLOS ONE’s publication criteria as it currently stands. Therefore, we invite you to submit a revised version of the manuscript that addresses the points raised during the review process.

We look forward to receiving your revised manuscript.

Kind regards,

Linton Munyai, PhD

Academic Editor

PLOS ONE

Journal Requirements:

Additional Editor Comments:

Good day Authors.

We have now received reports from our reviewers. Both reviewers provided comments and suggestion to improve the novelty of the manuscript. Most of the issues raised touches on methods, data analysis, results and discussion sections which requires serious attention. You will see that, while the reviewers find your work of interest, they have raised points that need to be addressed by a Major revision.

Reviewers' comments:

Reviewer's Responses to Questions

**Comments to the Author**

1. Is the manuscript technically sound, and do the data support the conclusions?

Reviewer #1: Yes

Reviewer #2: Partly

2. Has the statistical analysis been performed appropriately and rigorously?

Reviewer #1: Yes

Reviewer #2: Yes

3. Have the authors made all data underlying the findings in their manuscript fully available?

Reviewer #1: Yes

Reviewer #2: No

4. Is the manuscript presented in an intelligible fashion and written in standard English?

Reviewer #1: Yes

Reviewer #2: Yes

Reviewer #1: Review of the article “Effect of Long-Term Use of Treated Wastewater and Clean Water Mixtures on Soil Heavy Metal Accumulation: An Assessment Using Pollution Indices”

1. Introduction and Relevance of the Topic

The topic addressed by the author is highly relevant and socially significant in the context of current global challenges related to water and agriculture. The author accurately identifies an existing research gap. Most previous studies have focused on the short-term effects of wastewater reuse, while the long-term impacts of various mixing ratios with clean water are poorly understood. Moreover, the use of commonly applied pollution indices (CF, EF, Igeo, PLI) allows for a quantitative, comparable, and internationally recognized assessment of irrigation impact. The study also has strong application potential and may serve as a scientific basis for developing policies on wastewater and water resource management in agriculture, both at the local and national levels. The results may be particularly useful for decision-makers, agricultural planners, environmental protection agencies, and wastewater treatment plant operators. The author addresses an interdisciplinary topic of high environmental, agricultural, and societal relevance, and the study provides important data for sustainable water management in times of increasing pressure on natural resources.

2. Strengths of the Manuscript

• Topical relevance. The manuscript The paper addresses an important and current issue of treated wastewater reuse in agriculture in the context of increasing water scarcity.

• Correct and comprehensive methodology. The study employs four recognized pollution indices (CF, EF, Igeo, PLI) as well as appropriate analytical and statistical methods.

• Clear presentation of data. The results are well documented in tables and figures with clear statistical interpretation.

• Practical implications. The conclusions have applied value and may inform decisions regarding safe wastewater use in agriculture.

3. Errors and Ambiguities in the Text

Abstract

• Line 10 contains a sentence that is too long and stylistically unclear: "This study aimed to assess the effects of long-term irrigation using different ratios of treated wastewater (WW) and clean water (CW)...”. I suggest splitting this sentence into two or three logical segments.

• The term “long-term” is imprecise. The abstract does not explain what is meant by “long-term” and why a four-year period qualifies.

Introduction

• The objective of the study is not clearly stated.

• The author doesn't explicitly point out the research gap that prompted him to undertake this study. I suggest supplementing the text with something like: "Although numerous studies have examined the short-term effects of wastewater irrigation, the long-term impact on various water-to-water ratios remains under-researched."

Materials and Methods

• The pot-based experimental model is not sufficiently justified. The author (lines 105...) describes the use of pots in the study but does not discuss the limitations of this approach.

• There is no information on the mineral composition of the soil before the start of the experiment. Although the author mentions the soil classification, the initial properties (pH, SOM, EC, particle size distribution) are not provided. I recommend supplementing this section with a table of initial soil parameters.

• The author does not clarify what is meant by “long-term” or justify why a four-year period meets this criterion.

Results and Discussion

• The manuscript does not reference weather conditions during the experiment. Although general climatic characteristics are provided, there is no analysis of seasonal variability (precipitation, temperature) and its impact on the results.

• No information is provided on the metal content in plant tissues. The results section completely omits the aspect of bioaccumulation. I suggest adding a note that plants were not analyzed and recommending this as a direction for future research.

• There is no “Limitations” section. The discussion does not include any reflection on the limitations of the experimental design (e.g., pot scale, lack of plant analysis). I recommend adding a paragraph at the end of the discussion, for example: “The experimental design using pots may not fully reflect field-scale interactions, and the lack of plant uptake analysis limits the assessment of potential risks to the food chain.”

• The conclusions are too general. They do not indicate a specific safe range for irrigation. I suggest adding, for example: “Based on the applied pollution indices, irrigation with 25–50% treated wastewater appears environmentally safe over a four-year period.”

• It is necessary to check for missing spaces between words, e.g., in lines 138 and 168.

• Terminology should be standardized, e.g., “mg/kg” vs. “mg kg⁻¹”.

Conclusions

• The final conclusions are too general. No practical recommendations are provided, and there is no reference to possible long-term effects.

4. Language, style and structure

The language of the manuscript is generally correct but requires moderate editorial revision. The text contains overly long and complex sentences, especially in the abstract and discussion, which reduces readability. The style is at times too general, with imprecise use of terms like “effective” or “beneficial” without reference to specific data. Simplifying sentence structure and reducing repetition is recommended. The overall structure of the article is logical.

5. Summary

The article provides a valuable contribution to the scientific literature on the impact of treated wastewater on heavy metal accumulation in soil in the context of its long-term use in agriculture. The presented results are original and important for sustainable water resource management. However, the current version of the manuscript requires revision and clarification, particularly in terms of data presentation and interpretation, as well as consistency in language and structure.

Reviewer #2: The manuscript addresses an important and timely issue related to the sustainable use of treated wastewater for irrigation, with a focus on heavy metal accumulation in soils. The study is well-structured, employs recognized pollution indices (CF, EF, Igeo, PLI), and provides valuable data from a 4-year controlled experiment. However, some methodological details are insufficiently described, and the novelty compared to existing literature is not fully emphasized. The study would benefit from a stronger discussion linking results to broader environmental and human health risks. Overall, the paper is potentially suitable for publication after major revisions.

Comments:

1. Similar studies exist in different regions (Tunisia, Morocco, China, India). The authors should highlight more clearly what is novel in this study (e.g., the 4-year duration, proportions of wastewater/clean water, or the specific regional context).

2. The use of pots rather than field plots is a limitation; this should be clearly acknowledged and discussed. Results from pots may not fully reflect heavy metal dynamics in field soils.

3. Please clarify the sampling design: how many soil samples per pot/year were collected, and how were composite samples prepared?

4. The choice of background concentrations for CF, EF, and Igeo calculations needs justification. Were local background values determined, or were global averages used? This choice critically affects interpretation of contamination levels.

5. Language and Style

The manuscript is understandable but requires English editing for grammar, sentence structure, and conciseness.

Standardize terminology: use consistently “treated wastewater (WW)” and “clean water (CW)” (sometimes “fresh water” is mentioned).

6. References

Ensure uniform formatting of references (some include DOI, others do not).

Update citations with recent international standards and WHO guidelines where relevant.

7. Figures and Tables

Please include units in all axes and table headings.

Some tables (e.g., Table 6) are very dense. Consider summarizing key findings graphically.

8. Results and Discussion

Figures presenting CF, EF, Igeo, and PLI require improved readability (larger fonts, clearer legends, units on axes).

The discussion focuses mainly on soil pollution indices; however, a stronger link to potential crop uptake and human health implications would significantly improve the manuscript.

Please integrate comparisons with more regional studies (e.g., Hidri et al., 2014; Mkhinini et al., 2020), which address long-term wastewater use in Mediterranean contexts.

9. Conclusion

The conclusion is currently descriptive. It should be more critical and forward-looking, outlining clear recommendations for policymakers and practitioners (e.g., safe proportions of wastewater, crop choices, long-term monitoring strategies).

10. Ethics and Data Availability

The manuscript states that data are available “upon request.” PLOS ONE requires full data availability at submission. Authors should provide datasets as supplementary files or in a public repository.

**Do you want your identity to be public for this peer review?** For information about this choice, including consent withdrawal, please see our Privacy Policy

Reviewer #1: **Yes: ** Marcin Sidoruk

Reviewer #2: **Yes: ** Samir Ghannem

---

## [Author Response · Author response to Decision Letter 1]

9 Oct 2025

Dear Editor,

First of all, I would like to thank you and the reviewers for taking the time to evaluate our manuscript. We appreciate the constructive feedback and insightful comments provided, which have significantly helped us improve the quality of our work.

Below, we have provided detailed, point-by-point responses to the reviewers' comments and have made the necessary revisions to the manuscript as follows:

Journal Requirements:

1- Comment: Please ensure that your manuscript meets PLOS ONE's style requirements, including those for file naming.

Response: Thank you for your valuable guidance. The manuscript has been revised in accordance with the PLOS ONE style requirements, including formatting and file naming conventions, as recommended.

2- Comment: Please verify that all necessary raw data are provided so that the results of your study can be replicated, or upload it to a suitable data repository, obtain a doi number, and share the link.

Response: Thank you for highlighting the importance of transparent data availability. As recommended, the complete dataset required to replicate the study's findings — including raw values used in statistical analyses, figures, and tables — has been uploaded to Figshare, a stable public repository. The dataset is now openly accessible to all readers.

We have also updated the Data Availability Statement in the manuscript as follows:

“All datasets generated for this study can be found in Figshare (https://doi.org/10.6084/m9.figshare.30209449).”

We trust that this revision satisfies the journal’s data sharing requirements. Please let us know if any further clarification or formatting is needed.

3- Comment: If the reviewer comments include a recommendation to cite specific previously published works, please review and evaluate these publications to determine whether they are relevant and should be cited. There is no requirement to cite these works unless the editor has indicated otherwise.

Response: Thank you for your suggestion. We have carefully reviewed the previously published works recommended by the reviewers. After evaluating their relevance and contribution to our study, we have cited the appropriate ones in the revised manuscript where applicable. We appreciate the reviewers' insights, which have helped strengthen the context and depth of our discussion.

Reviewer 1

1- Comment: Line 10 contains a sentence that is too long and stylistically unclear: "This study aimed to assess the effects of long-term irrigation using different ratios of treated wastewater (WW) and clean water (CW)...”. I suggest splitting this sentence into two or three logical segments.

Response: Thank you for your valuable feedback. In accordance with your suggestion, we have revised the sentence in Line 10 to improve its clarity and readability. The revised sentence has been split into two logical parts to enhance stylistic flow.

2- Comment: The term “long-term” is imprecise. The abstract does not explain what is meant by “long-term” and why a four-year period qualifies.

Response: We appreciate the reviewer’s valuable observation regarding the ambiguity of the term “long-term.” To avoid confusion, we have revised the expression to “four-year field experiment” throughout the manuscript, particularly in the abstract and relevant sections. We believe this change provides a clearer and more precise description of the study duration.

3- Comment: The objective of the study is not clearly stated.

Response: We appreciate the reviewer’s comment regarding the clarity of the study's objective. In response, we have revised the aim statement in the Introduction section to explicitly define the scope, duration, and methodology of the research. The revised objective now reads:

"The primary aim of this study is to investigate how 4-year irrigation using different ratios of treated wastewater (WW) and clean water (CW) affects the accumulation of heavy metals (Cr, Ni, Cu, Zn, Cd, Fe, and Pb) in agricultural soil. Specifically, five irrigation treatments—100% WW, 75% WW + 25% CW, 50% WW + 50% CW, 25% WW + 75% CW, and 100% CW (control)—were applied over four consecutive years. The accumulation levels were evaluated using four widely accepted pollution indices: Contamination Factor (CF), Enrichment Factor (EF), Geoaccumulation Index (Igeo), and Pollution Load Index (PLI). Additionally, the study aims to assess the environmental risks and potential benefits of reusing treated wastewater in agriculture, with a focus on sustainable soil management and safe irrigation practices."

We hope this revision clarifies the research objective and aligns better with the journal's expectations.

4- Comment: The author doesn't explicitly point out the research gap that prompted him to undertake this study. I suggest supplementing the text with something like: "Although numerous studies have examined the short-term effects of wastewater irrigation, the long-term impact on various water-to-water ratios remains under-researched."

Response: Thank you for this valuable suggestion. We agree with the reviewer that highlighting the research gap more clearly improves the contextual clarity and significance of the study. Accordingly, we have revised the Introduction section and added the following sentence to better emphasize the gap in the existing literature:

"Although numerous studies have examined the short-term effects of wastewater irrigation, the long-term impact of different treated wastewater and clean water ratios on soil heavy metal accumulation remains under-researched, particularly when assessed through multiple pollution indices."

This sentence now clarifies the specific research gap our study aims to address.

5- Comment: The pot-based experimental model is not sufficiently justified. The author (lines 105...) describes the use of pots in the study but does not discuss the limitations of this approach

Response: We thank the reviewer for this important observation. We agree that while pot-based experiments offer practical advantages in controlling treatment applications and minimizing environmental variability, they also come with certain limitations, such as restricted root development, reduced soil buffering capacity, and limited representation of field-scale dynamics.

To address this concern, we have revised the Materials and Methods section to include a justification for using a pot-based model and a brief discussion of its limitations. The following paragraph has been added:

“A pot-based experimental model was selected in this study to ensure precise control over the irrigation regimes, particularly the ratio of treated wastewater to clean water applied over a multi-year period. This approach enabled consistent treatment application, minimized external environmental variability, and allowed systematic sampling of soil for heavy metal analysis over four consecutive years. However, we acknowledge that pot experiments may not fully simulate natural field-scale hydrological processes, such as deep percolation, runoff, and soil heterogeneity. Therefore, the results should be interpreted with consideration of these limitations, and further field-based research is recommended to validate the observed heavy metal accumulation patterns under real agricultural conditions.”

6- Comment: There is no information on the mineral composition of the soil before the start of the experiment. Although the author mentions the soil classification, the initial properties (pH, SOM, EC, particle size distribution) are not provided. I recommend supplementing this section with a table of initial soil parameters.

Response: Thank you for your valuable suggestion. In response to your comment, we have added detailed information regarding the initial physical and chemical properties of the experimental soil. These include particle size distribution, soil texture, electrical conductivity (EC), pH, CaCO₃, soil organic matter (SOM), total nitrogen (N), available phosphorus (P₂O₅), potassium (K₂O), exchangeable cations (Ca, Mg, K, Na), cation exchange capacity (CEC), extractable Na, ESP, and micronutrients such as B, Cu, Mn, and Zn. This information has been presented in the revised manuscript as Table 1 under the Materials and Methods section.

We hope this revision addresses your concern thoroughly.

7- Comment: The author does not clarify what is meant by “long-term” or justify why a four-year period meets this criterion.

Response: We appreciate the reviewer’s valuable observation regarding the ambiguity of the term “long-term.” To avoid confusion, we have revised the expression to “four-year field experiment” throughout the manuscript, particularly in the abstract and relevant sections. We believe this change provides a clearer and more precise description of the study duration.

8- Comment: The manuscript does not reference weather conditions during the experiment. Although general climatic characteristics are provided, there is no analysis of seasonal variability (precipitation, temperature) and its impact on the results.

Response: In response to the reviewer’s comment, additional details regarding the climatic conditions during the experimental period were included in the revised manuscript. Specifically, long-term meteorological data (1961–2024) and experimental-year data (2021–2024) for Bingöl Province were analyzed to assess seasonal variability in temperature and precipitation. The revised section now states that May–September represents the main vegetation period, characterized by a continental climate with hot and dry summers and rainfall concentrated at the beginning of the season (Fig. 1). Mean monthly temperatures during the experimental years (21.3–28.4°C) were consistent with long-term averages (21.3–26.7°C), and precipitation varied slightly between 6–40 mm compared to the long-term average of 49.2 mm. Although 2022 and 2023 received relatively higher early-season rainfall, 2021 and 2024 were drier, particularly in mid-summer. Relative humidity ranged from 26.4% to 47.8%, slightly below the long-term average (36–44%). Overall, both temperature and precipitation trends during the study period closely followed the regional long-term pattern. Therefore, it is considered that the experimental outcomes were not significantly influenced by seasonal variability, as climatic conditions remained relatively stable across the study years. (Experimental area and climate section)

9- Comment: No information is provided on the metal content in plant tissues. The results section completely omits the aspect of bioaccumulation. I suggest adding a note that plants were not analyzed and recommending this as a direction for future research.

Response: We appreciate the reviewer’s insightful comment regarding the omission of heavy metal content in plant tissues. As correctly noted, plant uptake and bioaccumulation were beyond the scope of this soil-focused study. The primary objective was to monitor soil contamination trends under long-term irrigation scenarios. However, we agree that investigating metal accumulation in crops would provide valuable complementary insights into the environmental and health implications of treated wastewater reuse. A statement acknowledging this limitation and recommending future research has been added to the conclusion section “Although this study focused solely on the accumulation of heavy metals in soil, it did not include an analysis of metal uptake by plants. Future research should investigate the potential bioaccumulation of these metals in plant tissues to better assess the food safety and ecological risks associated with the long-term use of treated wastewater in agricultural irrigation.”).

10- Comment: There is no “Limitations” section. The discussion does not include any reflection on the limitations of the experimental design (e.g., pot scale, lack of plant analysis). I recommend adding a paragraph at the end of the discussion, for example: “The experimental design using pots may not fully reflect field-scale interactions, and the lack of plant uptake analysis limits the assessment of potential risks to the food chain.”

Response: We thank the reviewer for this valuable suggestion. We fully agree that acknowledging the limitations of the study improves the scientific rigor and transparency of our work. In line with the reviewer’s recommendation, we have added a “Limitations” paragraph at the end of the Discussion section, highlighting the constraints of the pot-based experimental setup and the absence of plant uptake analysis.

“This study was conducted using a pot-based experimental setup, which provided controlled conditions for irrigation treatments but may not fully replicate field-scale hydrological processes such as deep percolation, lateral flow, or complex soil-plant interactions. Therefore, the results should be interpreted with caution when extrapolating to actual field conditions. In addition, the study focused exclusively on the accumulation of heavy metals in the soil, without analyzing their uptake and translocation in plant tissues. This limits our ability to assess potential health and ecological risks related to bioaccumulation through the food chain. Future research should incorporate field-scale experiments and include plant tissue analysis to provide a more comprehensive evaluation of the sustainability and safety of treated wastewater reuse in agriculture.”

11- Comment: “The conclusions are too general. They do not indicate a specific safe range for irrigation. I suggest adding, for example: ‘Based on the applied pollution indices, irrigation with 25–50% treated wastewater appears environmentally safe over a four-year period.’

Response: Thank you for the constructive suggestion. In response, we have revised the Conclusion section to include a more specific statement regarding the environmentally safe range of treated wastewater use. Based on the results of CF, EF, Igeo, and PLI indices, we now clearly state that the use of 25–50% treated wastewater mixtures for irrigation appears to be environmentally safe over the four-year period examined in this study.

“Based on the applied pollution indices (CF, EF, Igeo, and PLI), irrigation with 25–50% treated wastewater mixtures can be considered environmentally safe over a four-year period, as heavy metal accumulation remained below national and international contamination thresholds.”

12- Comment: It is necessary to check for missing spaces between words, e.g., in lines 138 and 168.”

Response: Thank you for pointing this out. The entire manuscript has been carefully rechecked and revised to correct all formatting issues, including missing spaces between words and other typographical inconsistencies.

13- Comment: “Terminology should be standardized, e.g., ‘mg/kg’ vs. ‘mg kg⁻¹’.”

Response: Thank you for your observation. All units and scientific notations throughout the manuscript have been revised and standardized to follow a consistent format, with “mg kg⁻¹” adopted uniformly as per scientific conventions.

14- Comment: The final conclusions are too general. No practical recommendations are provided, and there is no reference to possible long-term effects.

Response: Thank you for this insightful comment. In response, we have revised the conclusion section to include specific practical recommendations based on the findings. We now clearly state that irrigation with 25–50% treated wastewater mixed with clean water can be considered environmentally safe over a four-year period, according to the applied pollution indices. Additionally, we have discussed the potential long-term risks of heavy metal accumulation beyond the four-year scope of the study and emphasized the importance of soil monitoring and sustainable irrigation strategies. These additions aim to strengthen the practical applicability and long-term relevance of our research outcomes. The revised paragraph has been added to the Conclusion section.

15- Comment: The language of the manuscript is generally correct but requires moderate editorial revision. The text contains overly long and complex sentences, especially in the abstract and discussion, which reduces readability. The style is at times too general, with imprecise use of terms like “effective” or “beneficial” without reference to specific data. Simplifying s

---

## [Decision Letter · Decision Letter 1]

29 Oct 2025

Dear Dr. Er,

Thank you for submitting your manuscript to PLOS ONE. After careful consideration, we feel that it has merit but does not fully meet PLOS ONE’s publication criteria as it currently stands. Therefore, we invite you to submit a revised version of the manuscript that addresses the points raised during the review process.

We look forward to receiving your revised manuscript.

Kind regards,

Linton Munyai, PhD

Academic Editor

PLOS ONE

Journal Requirements:

**Additional Editor Comments:**

Your manuscript, "Effect of long-term use of treated wastewater and clean water mixtures on soil heavy metal accumulation: an assessment using pollution indices", has now been assessed. You will see that, while the reviewers find your work of interest, they have raised points that need to be addressed by minor revision.

Reviewers' comments:

Reviewer's Responses to Questions

**Comments to the Author**

Reviewer #2: (No Response)

2. Is the manuscript technically sound, and do the data support the conclusions?

Reviewer #2: Partly

3. Has the statistical analysis been performed appropriately and rigorously?

Reviewer #2: Yes

4. Have the authors made all data underlying the findings in their manuscript fully available?

Reviewer #2: Yes

5. Is the manuscript presented in an intelligible fashion and written in standard English?

Reviewer #2: Yes

Reviewer #2: The manuscript entitled “Effect of long-term use of treated wastewater and clean water mixtures on soil heavy metal accumulation: an assessment using pollution indices” presents a relevant and timely study addressing the environmental impacts of wastewater reuse in agriculture. The topic is significant for regions facing water scarcity, and the four-year experimental design adds valuable long-term insight.

Overall, the study is technically sound, well-structured, and written in clear English. The use of multiple pollution indices (CF, EF, Igeo, and PLI) provides a comprehensive assessment of soil contamination levels, and the results generally support the conclusions. The findings indicating that 25–50% treated wastewater mixtures can be safely used for irrigation are both practical and environmentally meaningful.

However, several points should be considered to strengthen the manuscript:

Experimental limitations: The pot-based experimental design does not fully replicate field-scale hydrological and biological processes. This limitation is acknowledged, but it should be emphasized more clearly in the Discussion and Conclusion sections.

Statistical reporting: The statistical analysis is appropriate but should be reported in greater detail (e.g., include F-values, degrees of freedom, and standard error bars in figures).

Lack of plant uptake data: Since only soil data were analyzed, a short discussion on potential bioaccumulation in crops and implications for food safety would improve the ecological relevance of the work.

Discussion depth: Strengthen the discussion by including mechanistic explanations of metal behavior (adsorption, mobility, pH influence) and by comparing results with recent regional studies, especially from Mediterranean or semi-arid contexts.

Figures and tables: Improve figure readability (larger fonts, clear legends, consistent units) and, if possible, include a location map of the study site.

References: Ensure uniform formatting and inclusion of DOIs for all references. Adding more recent studies (2020–2024) would further support the discussion.

In conclusion, the manuscript is of good scientific quality and makes a valuable contribution to sustainable wastewater management research. With minor revisions focused on improving clarity, discussion depth, and figure presentation, it will be suitable for publication.

**Do you want your identity to be public for this peer review?** For information about this choice, including consent withdrawal, please see our Privacy Policy

Reviewer #2: **Yes: ** I confirm that I am the sole author of this review and that I have not written it on behalf of another person.

---

## [Author Response · Author response to Decision Letter 2]

3 Nov 2025

Dear Editor,

First of all, I would like to thank you and the reviewers for taking the time to evaluate our manuscript. We appreciate the constructive feedback and insightful comments provided, which have significantly helped us improve the quality of our work.

Below, we have provided detailed, point-by-point responses to the reviewers' comments and have made the necessary revisions to the manuscript as follows:

Reviewer

1- Comment: The pot-based experimental design does not fully replicate field-scale hydrological and biological processes. This limitation is acknowledged, but it should be emphasized more clearly in the Discussion and Conclusion sections.

Response: We appreciate the reviewer’s insightful comment regarding the limitations of the pot-based experimental design. In response, we have revised the Discussion and Conclusion sections to more explicitly emphasize that while the pot-based approach allowed for consistent and controlled irrigation treatments over multiple years, it may not fully replicate field-scale hydrological and biological processes such as deep percolation, lateral water movement, or soil-plant interactions.

Accordingly, we now stress that the results should be interpreted with caution when extrapolating to real agricultural settings and have recommended that future studies employ field-scale designs to validate and expand upon our findings.

The following sentences has been added to the Discussion and Conclusion section to address this more clearly:

Discussion: “This study was conducted using a pot-based experimental setup, which offered a high level of control over irrigation regimes—particularly the treated wastewater and clean water ratios—and allowed for systematic and repeated soil sampling over four consecutive years. However, it is important to emphasize that this approach does not fully replicate field-scale hydrological and biological processes, such as deep percolation, surface runoff, soil heterogeneity, and complex root-soil interactions that naturally occur under open field conditions. These factors can significantly influence the mobility, retention, and bioavailability of heavy metals in soil. Additionally, the study focused solely on the accumulation of heavy metals in the soil compartment, without analyzing their uptake and translocation in plant tissues. This represents a limitation in fully assessing the potential human and ecological risks related to bioaccumulation through the food chain. Therefore, while the findings offer valuable insights under controlled conditions, caution must be exercised when extrapolating these results to real-world agricultural systems. Future research should involve long-term, field-based trials that incorporate both soil and plant analyses to more comprehensively evaluate the environmental sustainability, crop safety, and potential health implications of treated wastewater reuse in agriculture.” (page 15, Line 156-169).

Conclusion: “Although the findings of this study provide valuable insights into the dynamics of heavy metal accumulation under controlled irrigation with treated wastewater, it must be emphasized that the pot-based experimental design does not fully replicate field-scale hydrological and biological conditions. Therefore, the conclusions drawn here should be interpreted with caution, and future research should focus on long-term field experiments to validate these results under real agricultural conditions.” (page 16, Line 192-197).

2- Comment: Statistical reporting: The statistical analysis is appropriate but should be reported in greater detail (e.g., include F-values, degrees of freedom, and standard error bars in figures).

Response: Thank you for this valuable suggestion. Detailed statistical results are presented in Table 6 to enhance the transparency and clarity of the analysis. In accordance with the reviewer’s recommendation, Figure 6 has been revised to include standard error bars and significance annotations. Different letters now indicate statistically significant differences among treatments (p < 0.05), providing a clearer representation of the results.

3- Comment: Lack of plant uptake data: Since only soil data were analyzed, a short discussion on potential bioaccumulation in crops and implications for food safety would improve the ecological relevance of the work.

Response: We appreciate the reviewer’s insightful comment regarding the importance of plant uptake data and its implications for food safety. Indeed, understanding the potential transfer of heavy metals from soil to edible plant parts is essential for a comprehensive environmental risk assessment. Although this study focused solely on soil contamination and pollution index evaluations, we agree that including data on heavy metal accumulation in plant tissues would enhance the ecological relevance and public health perspective of the work. In response to this valuable suggestion, we have added a new paragraph in the Levels of heavy metals in soil section (page 12, Line 44-48; 64-74) that outlines relevant findings from the literature regarding bioaccumulation in crops irrigated with wastewater. This paragraph discusses possible health risks, the role of long-term exposure, and emphasizes the need for future studies integrating plant tissue analysis and dietary risk assessments. We believe this addition strengthens the manuscript’s context and supports the need for multidisciplinary approaches in sustainable wastewater reuse practices.

4- Comment: Discussion depth: Strengthen the discussion by including mechanistic explanations of metal behavior (adsorption, mobility, pH influence) and by comparing results with recent regional studies, especially from Mediterranean or semi-arid contexts.

Response: Thank you for this valuable suggestion. In the revised manuscript, we have significantly expanded the Discussion section by integrating mechanistic insights into the behavior of heavy metals under wastewater irrigation. Specifically, we addressed how key soil properties—such as pH, electrical conductivity (EC), cation exchange capacity (CEC), and organic matter (OM)—influence metal adsorption, mobility, and bioavailability. These explanations are now contextualized with current findings from Mediterranean and semi-arid environments (e.g., Batool et al., 2022; Laptiev et al., 2024; Dogan Demir and Sahin, 2020; Yerli et al., 2025). The revised discussion better links our observed changes in pollution indices with pH- and salinity-driven dynamics and provides a stronger scientific basis for interpreting the behavior of heavy metals in the studied soil systems. We believe this revision considerably enhances the ecological relevance and regional comparability of our findings. (page 8-9, Line 214-226).

5- Comment: Figures and tables: Improve figure readability (larger fonts, clear legends, consistent units) and, if possible, include a location map of the study site.

Response: Thank you for your constructive feedback. The figure quality has been enhanced to improve readability, including increased resolution and font sizes. All abbreviations used in the figures have been clearly defined in the legend sections for clarity and consistency. Additionally, a location map illustrating the national and regional context of the study area has been added as Figure 1. Map of the study area.

6- Comment: Ensure uniform formatting and inclusion of DOIs for all references. Adding more recent studies (2020–2024) would further support the discussion.

Response: Thank you for your helpful comment regarding the reference list. In response, we have thoroughly reviewed and revised all references to ensure consistent formatting and included DOI numbers where available. Additionally, to enhance the scientific depth and currency of the discussion, we have incorporated several recent studies published between 2020 and 2024, as recommended. These include: Ahmed et al. (2023), Batool et al. (2022), Ghamarnia et al. (2023), Laptiev et al. (2024), Shi et al. (2020), Tan et al. (2022), Tolossa (2021), Yang et al. (2021), Zhang et al. (2022), Villa et al. (2021), Yerli et al. (2025), Dogan Demir and Sahin (2020), Cakmakci and Sahin (2021), and Slima and Ahmed (2020). We believe these updates significantly strengthen the manuscript's relevance and alignment with current literature.

Thank you once again for your valuable guidance and support. We believe that the changes we have made will meet the reviewers' expectations and improve the manuscript's contribution to the field. We look forward to your feedback.

Sincerely,

Assist. Prof. Dr. Hasan ER

---

## [Editor Report · Decision Letter 2]

10 Nov 2025

Effect of long-term use of treated wastewater and clean water mixtures on soil heavy metal accumulation: an assessment using pollution indices

PONE-D-25-39184R2

Dear Dr. Er,

We’re pleased to inform you that your manuscript has been judged scientifically suitable for publication and will be formally accepted for publication once it meets all outstanding technical requirements.

Kind regards,

Linton Munyai, PhD

Academic Editor

PLOS ONE

---

## [Editor Report · Acceptance letter]

PONE-D-25-39184R2

PLOS ONE

Dear Dr. Er,

I'm pleased to inform you that your manuscript has been deemed suitable for publication in PLOS ONE. Congratulations! Your manuscript is now being handed over to our production team.

Kind regards,

on behalf of

Dr. Linton Munyai

Academic Editor

PLOS ONE